# Case Ascertainment on Australian Registers for Acute Rheumatic Fever and Rheumatic Heart Disease

**DOI:** 10.3390/ijerph17155505

**Published:** 2020-07-30

**Authors:** Treasure Agenson, Judith M. Katzenellenbogen, Rebecca Seth, Karen Dempsey, Mellise Anderson, Vicki Wade, Daniela Bond-Smith

**Affiliations:** 1School of Population and Global Health, The University of Western Australia, Perth 6009, Australia; 21960379@student.uwa.edu.au (T.A.); judith.katzenellenbogen@uwa.edu.au (J.M.K.); rebecca.cunneen@uwa.edu.au (R.S.); 2Telethon Kids Institute, Perth 6009, Australia; 3Menzies School of Health Research, Charles Darwin University, Darwin 0810, Australia; karen.dempsey@menzies.edu.au (K.D.); vicki.wade@menzies.edu.au (V.W.); 4Queensland Health, Brisbane 4000, Australia; mellise.anderson@health.qld.gov.au

**Keywords:** acute rheumatic fever, administrative data, epidemiology, linked data, disease register, rheumatic heart disease

## Abstract

In Australia, disease registers for acute rheumatic fever (ARF) and rheumatic heart disease (RHD) were previously established to facilitate disease surveillance and control, yet little is known about the extent of case-ascertainment. We compared ARF/RHD case ascertainment based on Australian ARF/RHD register records with administrative hospital data from the Northern Territory (NT), South Australia (SA), Queensland (QLD) and Western Australia (WA) for cases 3–59 years of age. Agreement across data sources was compared for persons with an ARF episode or first-ever RHD diagnosis. ARF/RHD registers from the different jurisdictions were missing 26% of Indigenous hospitalised ARF/RHD cases overall (ranging 17–40% by jurisdiction) and 10% of non-Indigenous hospitalised ARF/RHD cases (3–28%). The proportion of hospitalised RHD cases (36%) was half the proportion of hospitalised ARF cases (70%) notified to the ARF/RHD registers. The registers were found to capture few RHD cases in metropolitan areas (SA Metro: 13%, QLD Metro: 35%, WA Metro: 14%). Indigenous status, older age, comorbidities, drug/alcohol abuse and disease severity were predictors of cases appearing in the hospital data only (*p* < 0.05); sex was not a determinant. This analysis confirms that there are biases associated with the epidemiological analysis of single sources of case ascertainment for ARF/RHD using Australian data.

## 1. Introduction

Rheumatic heart disease (RHD) is the most common cause of acquired heart disease in children globally [1]. It is a sequela of acute rheumatic fever (ARF), caused by an abnormal immunological response to a group A streptococcal (GAS) pharyngitis [2] or impetigo [3]. Spontaneous resolution of ARF symptoms occurs in most cases over weeks to months; however, 50–75% will progress to the chronic valvopathy of RHD [1]. Most will develop RHD as a result of recurrent episodes of ARF, however, valve damage can occur after a single episode [4]. In Australia, time to disease progression has been shown to vary. Some 35% of Indigenous cases progress to RHD within two years and 61% within ten years of their first ARF episode [5]. Although the incidence rates of ARF and RHD substantially decreased in high-income countries during the 20th century [6], there remains a significant burden of disease in disadvantaged minority populations, including Aboriginal and Torres Strait Islander (hereafter respectfully, Indigenous) populations in Australia who have some of the highest ARF/RHD rates globally [1,7].

Intramuscular benzathine penicillin G (BPG) is widely used as a secondary prophylaxis to prevent recurrences in ARF episodes and subsequent RHD [3,8]. Previous studies have demonstrated a decrease in the rates of ARF recurrences by 87% to 96% [9] and reductions in complications associated with RHD with the regular administration of BPG [1,10]. The World Health Organization (WHO) recommends that countries faced with high ARF/RHD rates have “adequate monitoring and surveillance, as an integrated component of national health systems responses” [2,11]; for an overview of the evolution of ARF/RHD registers globally, see [12].

In Australia, both ARF and RHD are notifiable conditions in the Northern Territory (NT), Western Australia (WA), South Australia (SA) and Queensland (QLD), with the timing of the introduction of notification varying in these jurisdictions. In the remaining states and territories, ARF and RHD are only partially or not notifiable. In addition, jurisdiction-based ARF/RHD registers were established independently and at different times across the NT (1997), WA (2009), QLD (2006), SA (2010) and New South Wales (NSW) (2015). This resulted in different operational definitions and underlying data structures [10,13]. In SA and NT, clinicians have the option to enter the data of patients diagnosed with ARF and RHD directly into the register, however, most opt to faxing and emailing the control programs [13]. In WA and QLD, clinicians do not have the ability to manually enter data into the registers, but complete a notification form instead [13]. This results in control program staff being heavily involved with manual data handling and case validation [13]. The Federal government has published two reports (2013 and 2017) summarizing data from the jurisdictional registers [14,15]. No integrated, national register exists.

According to the Australian guidelines for ARF and RHD diagnosis (a modification of the Jones and WHO criteria), both ARF and RHD are to be considered as differential diagnosis when evaluating Indigenous children and adolescents with cardiac symptoms, and hospitalisation is required for all ARF episodes as soon as possible after symptom onset [10]. RHD cases can be expected to be hospitalised for case management, especially for RHD-associated complications. Thus, in theory most cases identified through ARF/RHD register records should appear in hospital administrative data, and vice versa. In practice, gaps in case ascertainment on the ARF/RHD registers may persist and additional data is required to obtain more complete data for determining the ARF/RHD burden.

The END RHD in Australia: Study of Epidemiology (ERASE) Project aims to provide the first quasi-national epidemiological profile of ARF/RHD using linked data from multiple data sources. Linked administrative data allows for the follow-up of cases across health system contacts and facilities over time, avoiding overestimation of diagnoses and caseloads. The project works closely with the End RHD Centre for Research Excellence [16] which has developed a strategy to remove RHD as a public health problem in the country. ERASE has assembled a linked dataset covering five jurisdictions (NSW, NT, QLD, SA, WA) to facilitate ARF/RHD epidemiological research in Australia [17,18]. Pertinent to the current study, ARF and RHD case identification was based on probability-linked ARF/RHD register, hospital admission and death data.

The overall aim of this study is to evaluate case ascertainment in the Australian ARF/RHD registers by comparing register records to administrative hospital data. The specific objectives were to:(1)describe the characteristics of ARF/RHD cases ascertained in the ARF/RHD register data, hospital data and both sources,(2)quantify case ascertainment by the ARF/RHD registers by determining the proportion of hospitalised ARF/RHD cases found in the ARF/RHD register data,(3)investigate the agreement of ARF and RHD diagnosis dates recorded on ARF/RHD registers and in the hospital data, and(4)identify factors associated with hospitalised ARF/RHD cases not being notified and recorded on the ARF/RHD registers.

## 2. Materials and Methods

### 2.1. Study Design and Data Sources

This is a validation study of case ascertainment based on ARF/RHD diagnoses recorded in two different case ascertainment systems, namely the register and administrative hospital records. The parent ERASE database identified ARF and RHD cases diagnosed in paediatric and adult populations in NT, SA, QLD, NSW and WA between July 2001 and June 2017. These five jurisdictions are home to 86% of the Australian Indigenous population (at 30 June 2016) [19]. Cases diagnosed outside their jurisdiction of residence were excluded, because we do not have access to cross-jurisdictionally linked data except for NT and SA. For the current study, ARF or RHD diagnoses had to be recorded in either data source (administrative hospital records or register) at least one calendar year following the establishment of a register in their jurisdiction of residence (NT from July 2001, SA from January 2013, QLD from January 2010 and WA from January 2011). Due to the NSW’s register’s recent inception, NSW data was not included. The study excluded foreign residents and patients with ARF/RHD diagnoses made before register establishment or outside an individual’s jurisdiction of residence. Administrative hospital data included records from public and private hospitals (NT and SA public hospitals only).

An ARF episode was defined as an episode recorded on an ARF/RHD register or as the principal diagnosis in an admissions record (International Classification of Diseases 10th Edition Australian Modification (ICD-10-AM) code: I00–I02) [14]. A unique episode was defined as an ARF record >90 days from the previous one and the earliest available diagnosis date was used to define episode onset [20,21].

A person was defined to be an RHD case identified on an ARF/RHD register from the earliest date they had an RHD severity assessment evaluated as “mild”, “moderate” or “severe” or had an RHD-related surgery or procedure recorded. Cases from the hospital data were defined from the earliest admission date where a predictive algorithm developed by the ERASE project assigned a predictive probability of being a valid RHD case. The prediction model considered ICD-10-AM codes I05–I09 as well as demographic and clinical variables [22,23]. RHD was defined to be “severe” if an individual had a hospital discharge diagnosis of heart failure, a procedure code indicating a valvular procedure or surgery (see Table A1 in the Appendix A) or was evaluated by a cardiologist as having severe RHD and documented on an ARF/RHD register accordingly.

The sample contains individuals 3–59 years of age at time of diagnosis. Individuals 60 years and over were excluded from the study, as this algorithm was not validated for this age group [17,18]. Individuals under the age of three were excluded to reduce the risk of misclassification of congenital heart disease as RHD.

Other variables of interest included baseline demographic (age, sex, vital status, jurisdiction of residence, Accessibility/Remoteness Index of Australia (ARIA) distinguishing between “very remote”, “remote”, “outer regional”, “inner regional” and “metropolitan” areas, socioeconomic status by population quintile (SES)), clinical (diagnosis (ARF or RHD), disease severity (mild/moderate or severe), ARF episode number (first-ever or recurrence), comorbidities and complications of RHD) and ARF/RHD register-related (last ARF/RHD diagnosis since register establishment) variables. The Socioeconomic Indexes for Areas (SEIFA) rank areas in Australia based on relative socio-economic advantage and disadvantage. For this study, Indigenous cases were allocated an Indigenous-specific socioeconomic index (Indigenous Relative Socioeconomic Outcomes index, IRSEO). The study used the most recent available recordings of SEIFA, IRSEO, ARIA, age and jurisdiction of residence for each individual. As comorbidity indices such as the Charlson Comorbidity Index [24] are not suitable for childhood conditions [25], individual comorbidities based on ICD-10-AM coded hospital admissions were investigated. The comorbidities included chronic obstructive pulmonary diseases, chronic kidney disease, other cardiovascular diseases, coronary heart disease, anticoagulant treatment, diabetes and mental health conditions. The complications considered for the analysis were heart failure, stroke, endocarditis and atrial fibrillation. Drugs/alcohol abuse and pregnancy (post-ARF/RHD diagnosis) were also investigated (see Table A2 in the Appendix A).

### 2.2. Statistical Methods

Figure 1 provides a visual overview of the cohorts for each sub-analysis. Univariate comparisons of demographic, clinical and ARF/RHD register-related variables were conducted to describe the characteristics of cases identified through the two different sources of ARF/RHD records. We identified statistical significance at the 0.05 level using a two-sided Chi-squared test. To describe case ascertainment on the registers, we calculated the percentage of ARF/RHD patients identified in hospital administrative data who also had an ARF/RHD record on a register. We also calculated the percentage of diagnosis dates recorded on an ARF/RHD register that were the equal or prior to hospital admission dates to determine the agreement of case ascertainment systems regarding the earliest reliable indication of an ARF/RHD diagnosis of each case. Inferential analyses were conducted using multivariate logistic regression models. The outcome variable identified the source of the record (1 = hospital only, 0 = register only or both register and hospital). Model selection was based on a priori inclusion of important covariates (SES, age, sex, disease severity, jurisdiction) and a backward stepwise regression methodology. The final model excluded complications due to its collinearity with disease severity. All analyses were performed using RStudio (V1.1.463) RStudio PBC, Boston, MA., USA), and Microsoft Excel (Microsoft Corporation, Redmond, CA, USA).

## 3. Results

### 3.1. Descriptive Analysis (Objective 1)

Of the 7321 cases in either source (Figure 2), 5824 were Indigenous (80%) (Table 1). Women comprised ~60% of cases for all data sources for both Indigenous and non-Indigenous patients. The cases only found in the hospital data had a higher percentage of individuals with ‘RHD only’ diagnoses, comorbidities, complications and a history of drugs/alcohol abuse compared with those only recorded on a register (all *p* < 0.001). Hospital cases were also older (*p* < 0.001), with >60% being 35 years or older among Indigenous patients and >60% being 45 years or older for non-Indigenous patients. This compares with 47% being 24 years or younger and 56% being 34 years or younger for Indigenous and non-Indigenous register only cases. Non-Indigenous cases had more than double the proportion of individuals appearing in the hospital data only (76%) compared with Indigenous cases (31%).

#### 3.1.1. Indigenous Cases

The NT was the primary jurisdiction of residence for Indigenous cases (Table 1). Across all case ascertainment systems, Indigenous cases were predominantly from remote areas and from areas with the lowest SES (all *p* < 0.001). Cases from the two highest SES areas (Quintile I & II) were mostly found in the hospital data only. As social disadvantage increased, the proportion of individuals in the hospital data only appeared to decrease. There was little difference in the time of diagnosis relative to register establishment between cases only found in the hospital data and cases that were only register-recorded (40% versus 41% diagnosed within two years after register establishment, *p* < 0.001). 23% of Indigenous people in the hospital data also have a death record and accounted for 62% of all deceased Indigenous cases (Table 1). The majority of SA cases appeared in the hospital data only (47%) whereas the NT had the highest proportion of individuals appearing in both data sources (likely also related to the fact that NT has the largest number of cases overall). The characteristics of the cases in the hospital data only were similar when stratified by jurisdiction (see Table A3 in the Appendix A).

#### 3.1.2. Non-Indigenous Cases

QLD was the primary jurisdiction of residence for non-Indigenous cases (Table 1). The cases appearing in the hospital data only were predominantly from the most socioeconomically advantaged areas, while cases appearing on both the register and hospital data or the register record only, predominantly resided in the most socioeconomically disadvantaged areas (*p* < 0.001). Cases residing remotely or in more disadvantaged areas had the lowest proportion of individuals in the hospital data only (Table 1).

### 3.2. Case Ascertainment on ARF/RHD Registers (Objective 2)

For both Indigenous and non-Indigenous cases, the proportion of hospitalised ARF/RHD cases that were also recorded on a register was lower for RHD cases (45% Indigenous, 13% non-Indigenous) compared with ARF cases (75% Indigenous, 33% non-Indigenous) (Table 2).

#### 3.2.1. Indigenous Cases

QLD had 68% of ARF cases appearing on the register, and SA had only 21% of Indigenous cases with RHD included on the register (Table 2). The Northern areas of Australia (NT Top End: 46%, QLD North: 55%, WA North: 63%) had a higher percentage value of RHD cases appearing on the register compared with the rest of the country (NT Central: 37%, QLD Other: 38%, WA Other: 30%, SA Other: 33%), and especially with metropolitan areas (QLD Metro: 35%, WA Metro: 14%, SA Metro: 13%) (Figure 3).

#### 3.2.2. Non-Indigenous Cases

Non-Indigenous patients across all jurisdictions had less than 50% of hospitalised ARF cases and less than 25% of hospitalised RHD cases on the register (Table 2).

### 3.3. Agreement between Recorded Diagnosis Dates (Objective 3)

For both Indigenous and non-Indigenous cases, QLD and WA had >90% agreement between ARF episode dates recorded on their registers and hospital data (Table 3). In contrast, for Indigenous cases, NT and SA had <40% of ARF register episode dates on the registers agreeing with those found in hospital data and <15% for non-Indigenous episodes. The agreement in dates for RHD was higher for non-Indigenous cases compared with Indigenous cases (66% versus 60% across all jurisdictions).

Mean and median difference between dates recorded on the registers and hospital data were more variable among jurisdictions for RHD (mean: 79–347 days, median: 4–74 days) compared with ARF (mean: 3–15 days, median: 2–14 days). We also conducted a sensitivity analysis where we considered a 14-day ‘grace period’ for ARF onset and a 90-day ‘grace period’ for RHD onset. Any dates falling within these time windows across data sources were still considered to be in agreement. The agreement was greater than 89% for ARF and 77% for RHD (see Table A4 in the Appendix A).

### 3.4. Multivariate Inferential Analysis (Objective 4)

Non-Indigenous cases had 3.1 times higher odds of appearing in the hospital data only (95% CI 2.4–3.9, *p* < 0.001) after adjusting for age, sex, disease severity, vital status, jurisdiction, ARIA, SES, comorbidity, drugs/alcohol abuse and register establishment (Table 4). Older age, comorbidities, drugs/alcohol abuse and disease severity but not sex were determinants of register exclusion. As the age group of the cases increased the adjusted odds of appearing in the hospital data only increased considerably. After adjustment, as disease severity increased the odds of being in the hospital data only decreased. The adjusted odds ratios (AOR) were not statistically significant for ARIA, vital status, comorbidities and register establishment for non-Indigenous cases, and SES for Indigenous cases.

#### 3.4.1. Indigenous Cases

Indigenous cases in the 55–75 age bracket had 22 times (95% CI 15–32, *p* < 0.001) higher odds of appearing in the hospital data only compared with individuals in the 3–14 age bracket. Cases who were last diagnosed more than two years after register establishment had 2.9 times the odds of not appearing on the register (95% CI 2.2–3.9, *p* < 0.001). When compared with cases from NT, cases residing in QLD had lower odds of being in the hospital data only (AOR: 0.77 95% CI: 0.62–0.96, *p* = 0.018). Those with comorbidities were 1.3 times more likely to appear in hospital records only (95% CI: 1.1–1.5, *p* = 0.011).

#### 3.4.2. Non-Indigenous Cases

For non-Indigenous cases, those in the 55–75 age bracket had 31 times the odds of appearing in the hospital data only compared with those in the 3–14 age bracket (95% CI 14–72, *p* < 0.001). Social disadvantage increased the odds of inclusion in the register with the most disadvantaged areas being associated with the highest odds of inclusion (AOR: 0.29, 95% CI 0.17–0.49, *p* < 0.001). Compared with the NT, residence in SA, QLD or WA increased the odds of cases appearing in the hospital data only (varying AOR: 2.6–30, *p* < 0.05).

## 4. Discussion

Register-based ARF/RHD control programs are most useful when case ascertainment is high, and accurate, timely and complete information appears in the database [21,26,27]. This study presents the first quasi-national Australian analysis investigating case ascertainment of ARF and RHD cases through ARF/RHD registers and hospital records using linked administrative data. We found that no single source provides comprehensive case ascertainment by itself, with 31% and 26% of Indigenous and 76% and 10% of non-Indigenous cases missing from the register and hospital data respectively. Differences in patient characteristics captured through the two case ascertainment systems (particularly age, Indigenous status, disease severity and comorbidity profiles) highlight the potential for bias associated when only one data source is considered. Therefore, realistic epidemiological analyses and policy targets for ARF/RHD control cannot be developed using only hospital or register data, a limitation commonly faced in the previous literature. Analyses conducted using only hospital data will likely be biased towards older, comorbid cases with greater disease severity, residing in more metropolitan areas. If only register data is used, a bias towards a younger ARF/RHD population with lower disease severity and a lower socioeconomic profile would likely be incurred.

Twice as many ARF cases than RHD cases were found in both ARF/RHD register and hospital data. There was substantial variability between jurisdictions and regions in the reliability of case ascertainment through the ARF/RHD registers. The registers recorded proportionally more Indigenous cases from the Northern areas of Australia (NT Top End, QLD North, WA North) than non-Indigenous cases or individuals residing in other areas, especially metropolitan regions. The agreement between diagnosis dates across data sources also varied substantially by jurisdiction. For ARF, Queensland and WA performed substantially better than NT and SA.

We found that non-Indigenous status remains a significant predictor of being missed by the ARF/RHD registers of cases, after adjustment for age, sex, disease severity, vital status, jurisdiction, ARIA, SES, comorbidity, drugs/alcohol abuse and register establishment. Other significant determinants of exclusion from registers include older age, comorbidities, drugs/alcohol abuse and disease severity, but not sex.

Prior to this analysis, few studies have validated case ascertainment on ARF/RHD registers. Two Australian studies reported results on case ascertainment on the WA and NT ARF/RHD registers as incidental findings, rather than the primary study goal. In those small, regional studies, the WA and NT ARF/RHD registers were found to be incomplete (NT: 19% incomplete; Kimberley region of WA: 27% incomplete) [21,26] and containing errors due to the manual data entry process [13]. Studies in Fiji and New Zealand have also found gaps in case ascertainment on ARF/RHD registers [20,28].

The disparities between population groups and jurisdictions in case ascertainment observed in this study can be attributed to policy differences and operational considerations. All jurisdictions apart from SA, introduced ARF notification before the development of ARF/RHD disease registers, with rheumatic heart disease only becoming notifiable more recently. In SA, delayed notification requirements for ARF/RHD (in 2016 four years after the establishment of the ARF/RHD register) may have contributed to the lowest proportion of register notifications observed in this study. All jurisdictional registers mainly employ passive surveillance with the registration process relying on clinicians and health service providers being aware of ARF/RHD and their knowledge of having to complete notification forms [10]. Case ascertainment through the ARF/RHD register may be improved by more automated notification processes and standardised protocols for data entry; this would also reduce the labour-intensive need for manual data handling by the registers and the associated risks for data quality.

However, notification requirements can be complemented by other policies to increase case ascertainment. Outside of the NT, QLD had the highest proportion of hospitalised RHD cases recorded on its ARF/RHD register compared to the other jurisdictions despite RHD becoming notifiable outside of our research study period for this jurisdiction (2018). This may be related to comprehensive active case finding work in QLD that commenced in 2014 and identified 500 ARF/RHD cases previously unknown to the register [29], highlighting the importance of this approach to case ascertainment as part of control programs.

Operationally, the ARF/RHD registers face challenges such as limitations in resourcing with regard to infrastructure and staff. These operational constraints necessitate focusing on population groups and geographical areas where active case finding will likely identify a relatively large number of cases. It seems likely that such practical and clinically pertinent considerations have influenced the large representation of younger, Indigenous and Northern Australian cases recorded on ARF/RHD registers.

Patient transfers and low clinical awareness of ARF and RHD may have also lowered the case ascertainment by registers [10]. We found that metropolitan areas (generally better-resourced) had a relatively lower proportion of ARF/RHD patients on the registers compared with the rest of Australia (generally less-resourced). This suggests a need to increase clinical awareness and education of health service providers across Australia about ARF/RHD and notification requirements, including in metropolitan areas. The observed limitations in case ascertainment on the ARF/RHD registers and their likely causes point to the benefits of establishing a central, national ARF/RHD register based on automated notification and data management processes to achieve both more accurate epidemiological monitoring and more consistent real-time patient care including those across jurisdictional boundaries. Conversely, not all register recorded cases were found in the hospital data either. Despite guidelines recommending the hospitalisation of ARF patients [10], one-quarter (710 cases) of Indigenous ARF cases appeared in the register records only. As suggested by Artuso et al. [30], these cases were likely notified by primary health care services without hospitalisation due to a range of factors known to affect hospital utilisation by Indigenous patients including escort ineligibility/ availability, competing family priorities and mistrust of the health system.

For both case ascertainment methods, ARF/RHD register and hospital data, some cases may have been missed for various reasons. Sub-clinical ARF cases were likely under-represented due to a presumed high number of subclinical cases not seeking medical care, unavailability of primary health care data, underdiagnosis due to limited knowledge of ARF by clinicians and the clinical complexity associated with making an ARF diagnosis, including the lack of a definitive diagnostic test [10,31]. Miscoding in the hospital data (resulting in cases not being identified as ARF/RHD) and the limited study period may have also contributed to incomplete case ascertainment. Additionally, this study did not investigate case ascertainment for individuals last hospitalised for ARF/RHD before register establishment in their jurisdiction of residence, because the ARF/RHD registers do not generally engage in retrospective case finding. Current privacy limitations in cross-jurisdictional data linkage outside of SA and NT limited the tracking of patients across jurisdictions. It would be beneficial for a future study to investigate the characteristics of cases in primary health care services, as this data was not systematically available for this analysis. Small sample sizes, particularly among non-Indigenous cases, may have caused noise in the data and affected the power of the regression analysis. Furthermore, compared with cases appearing in the hospital data, our ability to describe the characteristics of the register-only cases was limited by the lack of recording of patient information relating to, for example, comorbidities, SES and remoteness. NSW was not included in our analysis, because its ARF/RHD register was only established at the end of 2016.

## 5. Conclusions

The NT, SA, QLD and WA ARF/RHD registers are part of the national Rheumatic Fever Strategy for control programs. Besides their surveillance function, an important goal of the register-based programs is to strengthen the prevention of ARF recurrences and minimise RHD progression by supporting the regular provision of secondary prophylaxis [13]. The effectiveness of the control programs is affected by incomplete case ascertainment. Pertinently, younger, Indigenous and Northern Australian cases are more comprehensively represented on ARF/RHD registers. However, differences in the characteristics of ARF/RHD cases across administrative hospital records and ARF/RHD registers demonstrate the need to utilise multiple sources when investigating the epidemiology of ARF/RHD in Australia to minimise systematic biases. Increased awareness of ARF/RHD in general and specifically of the notification requirements amongst clinicians would improve case ascertainment under the current operational systems. Furthermore, moving towards more integrated and automated systems, ideally implemented as a central, national ARF/RHD register, has the potential to improve communication and cooperation between the registers and health care services, minimise the workload of clinicians and the double handling of data and thus increase case ascertainment and data accuracy. This study demonstrates the need for sophisticated monitoring and surveillance systems in the global effort to reduce the burden of ARF and RHD.

## Figures and Tables

**Figure 1 ijerph-17-05505-f001:**
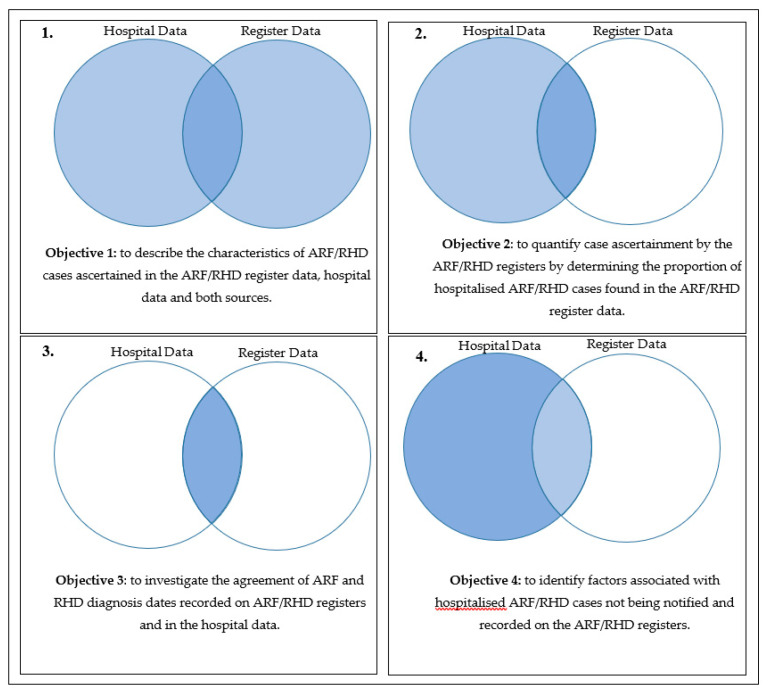
Study objectives and sub-samples.

**Figure 2 ijerph-17-05505-f002:**
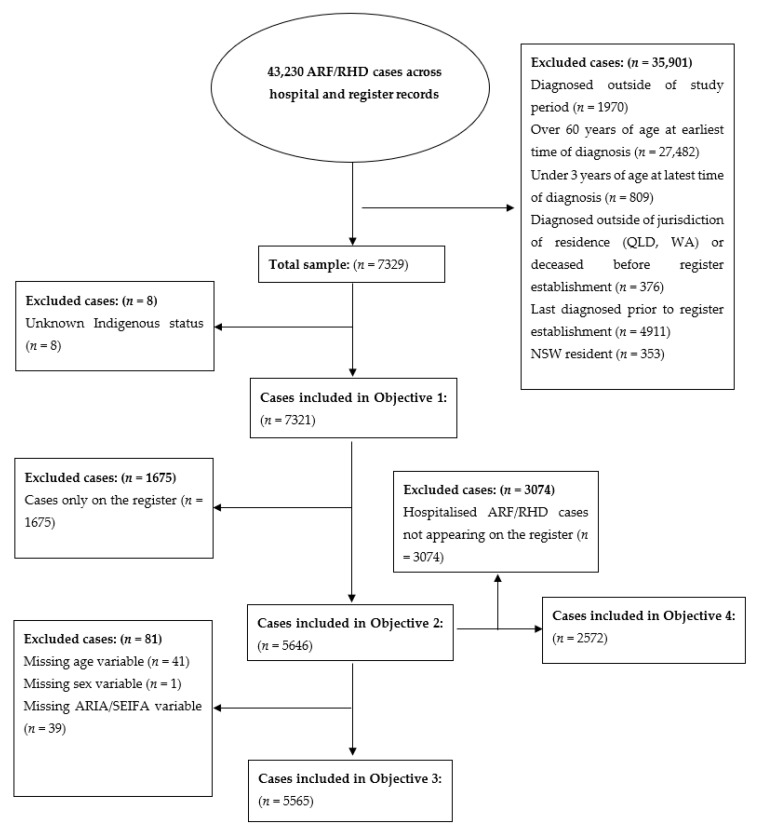
Flowchart of sample selection.

**Figure 3 ijerph-17-05505-f003:**
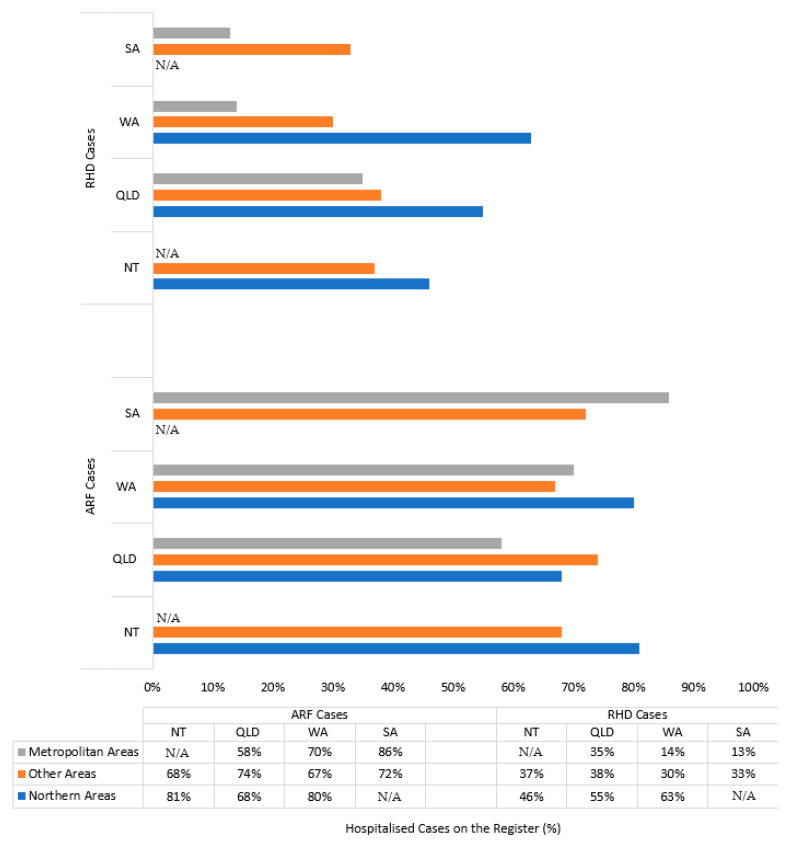
Percentage of hospital diagnosed acute rheumatic fever (ARF) and rheumatic heart disease (RHD) cases recorded on ARF/RHD registers for Indigenous cases by jurisdiction and geography.

**Table 1 ijerph-17-05505-t001:** Descriptive analysis of the study samples by Indigenous status and data sources.

Variable	Indigenous (*n* = 5824)							Non–Indigenous (*n* = 1497)						
	Hospital Only		Both		Register Only			Hospital Only		Both		Register Only		
	Cases *n*	R%	*n*	R%	*n*	R%	*p*-Value	*n*	R%	*n*	R%	*n*	R%	*p*-Value
Total	1821	31%	2484	43%	1519	26%		1135	76%	206	14%	156	10%	
Most Recent Age (years)							<0.001							<0.001
3–14	107	12%	535	59%	261	29%		35	38%	32	35%	24	26%	
15–24	159	12%	720	54%	457	34%		50	36%	46	33%	42	30%	
25–34	294	25%	548	46%	344	29%		90	67%	23	17%	22	16%	
35–44	441	46%	319	33%	209	22%		149	78%	28	15%	15	8%	
45–54	424	54%	225	28%	141	18%		262	83%	36	11%	18	6%	
55–75	393	62%	137	22%	107	17%		511	87%	41	7%	35	6%	
Sex							0.153							0.725
Male	653	30%	944	43%	592	27%		443	75%	80	14%	66	11%	
Female	1168	32%	1539	42%	927	26%		692	76%	126	14%	90	10%	
Diagnosis							<0.001							<0.001
ARF Only	284	22%	551	42%	463	36%		142	71%	28	14%	29	15%	
RHD Only	1457	47%	856	27%	809	26%		985	81%	119	10%	108	9%	
Both	80	6%	1077	77%	247	18%		8	9%	59	69%	19	22%	
Disease Status							<0.001							<0.001
Severe RHD	765	40%	911	48%	226	12%		740	80%	133	14%	48	5%	
Non-Severe RHD	772	29%	1022	39%	830	32%		253	67%	45	12%	79	21%	
ARF Only	284	22%	551	42%	463	36%		142	71%	28	14%	29	15%	
ARF Recurrence							<0.001							0.122
Total	365	14%	1628	60%	710	26%		150	53%	87	31%	48	17%	
No	313	14%	1222	55%	668	30%		145	53%	81	30%	48	18%	
Yes	51	10%	406	81%	42	8%		5	45%	6	55%	0	0%	
Vital Status							<0.001							0.056
Alive	1403	27%	2264	44%	1483	29%		1023	75%	193	14%	148	11%	
Dead	418	62%	220	33%	36	5%		112	84%	13	10%	8	6%	
State of Residence							<0.001							<0.001
NT	1207	36%	1580	47%	591	17%		81	52%	31	20%	43	28%	
SA	54	47%	33	28%	29	25%		135	91%	8	5%	6	4%	
QLD	380	24%	585	38%	593	38%		705	73%	163	17%	101	10%	
WA	180	23%	286	37%	306	40%		214	96%	<5	2%	6	3%	
Remoteness (ARIA)							<0.001							<0.001
Inner regional	164	54%	91	30%	50	16%		867	82%	136	13%	53	5%	
Outer Regional	341	35%	407	41%	235	24%		190	71%	49	18%	29	11%	
Remote	1303	31%	1982	47%	975	23%		55	59%	19	20%	20	21%	
Socioeconomic Status (SES)							<0.001							<0.001
Quintile I & II	245	42%	202	34%	141	24%		362	87%	31	7%	25	6%	
Quintile III	282	35%	343	43%	176	22%		248	78%	49	15%	21	7%	
Quintile IV	336	31%	442	40%	317	29%		221	79%	38	14%	19	7%	
Quintile V	945	31%	1493	49%	626	20%		283	70%	86	21%	36	9%	
Last Interaction Since Register Establishment (years)							<0.001							<0.001
Mean Time	6.7		9.6		6.3			3.5		5.3		5.5		
0–2	238	40%	113	19%	247	41%		295	84%	34	10%	21	6%	
2+	1583	30%	2371	45%	1272	24%		840	73%	172	15%	135	12%	
Comorbidity														
Any	1335	44%	1268	42%	415	14%	<0.001	873	84%	128	12%	38	4%	<0.001
Complications														
Any	769	54%	570	40%	93	6%	<0.001	601	85%	85	12%	22	3%	<0.001
Drug/Alcohol Abuse														
Any Substance	1240	43%	1124	39%	527	18%	<0.001	414	84%	51	10%	28	6%	<0.001
Pregnancy														
Females of Reproductive Age	747	30%	1095	43%	679	27%	<0.001	250	70%	64	18%	45	13%	0.004
Pregnancy	332	35%	480	50%	144	15%	<0.001	68	74%	15	16%	9	10%	0.540

**Table 2 ijerph-17-05505-t002:** Count and percentage of hospital diagnosed ARF/RHD cases recorded on ARF/RHD registers by Indigenous status and jurisdiction.

Indigenous Status		ARF			RHD			
		Total Hospital Cases	Hospital Cases on the Register (%)		Total Hospital Cases		Hospital Cases on the Register (%)	
Indigenous								
Jurisdiction	NT	1133	874	77%	2029	881	43%	
	SA	25	19	76%	66	14	21%	
	QLD	414	280	68%	622	319	51%	
	WA	226	172	76%	283	131	46%	
	Total	1798	1345	75%	3000	1345	45%	
Non-Indigenous								
Jurisdiction	NT	30	13	43%	87	19	22%	
	SA	8	<5	12%	135	7	5%	
	QLD	167	59	35%	721	116	16%	
	WA	27	<5	15%	193	<5	1%	
	Total	232	77	33%	1136	143	13%	
Indigenous & non-Indigenous		2030	1422	70%	4136		1488	36%

**Table 3 ijerph-17-05505-t003:** Agreement between recorded diagnosis dates for ARF and RHD cases on both sources by Indigenous status, jurisdiction and Indigenous region.

Indigenous Status		ARF						RHD					
		Episodes on Both Sources	Agreeing Date		Time Difference of Non-Agreeing Dates (Days)			Diagnosis on Both Sources	Agreeing Dates		Time Difference of Non-Agreeing Dates (Days)		
Indigenous		Total	*n*	%	Mean	Median	IQR	Total	*n*	%	Mean	Median	IQR
Jurisdiction	NT	979	396	40%	6	3	5	881	548	62%	347	8	324
	SA	19	6	32%	6	4	6	14	5	36%	92	5	5
	QLD	279	268	96%	15	14	22	319	163	51%	331	56	445
	WA	186	174	94%	3	2	2	131	93	71%	258	59	262
	Total	1463	844	58%	6	3	5	1345	809	60%	332	16	348
Indigenous Regions (IREGs)	NT Top End	733	306	42%	5	2	4	666	405	61%	298	6	187
	NT Central	246	90	37%	7	5	7	215	143	67%	526	17	609
	SA Other	13	<5	31%	7	5	5	9	<5	33%	24	3	3
	SA Metro	6	<5	33%	6	2	5	5	<5	40%	227	7	331
	QLD North	244	235	96%	15	14	18	273	133	49%	317	49	334
	QLD Other	23	22	96%	1	1	0	21	16	76%	724	532	1262
	QLD Metro	12	11	92%	27	27	0	25	14	56%	336	439	531
	WA North	135	130	96%	5	3	3	105	77	73%	231	54	226
	WA Other	32	28	88%	1	1	0	18	9	50%	372	218	580
	WA Metro	19	16	84%	2	2	1	8	7	88%	1	1	0
Non-Indigenous													
Jurisdiction	NT	13	<5	15%	7	5	7	19	13	68%	254	74	350
	SA	<5	0	0%	6	6	0	7	<5	57%	79	4	116
	QLD	58	56	97%	14	14	12	116	76	66%	239	6	141
	WA	<5	<5	100%				<5	<5	100%			
	Total	76	62	82%	8	6	7	143	94	66%	231	7	146

**Table 4 ijerph-17-05505-t004:** Multivariate inferential analysis of ARF and RHD cases only found in the hospital data.

Variable	Total (*n* = 5565)					Indigenous (*n* = 4284)					Non-Indigenous (*n* = 1281)				
	OR	(CI)	AOR	(CI)	*p* Value	OR	(CI)	AOR	(CI)	*p* Value	OR	(CI)	AOR	(CI)	*p* Value
Indigenous status															
Non-Indigenous	7.3	(6.2–8.5)	3.1	(2.4–3.9)	<0.001										
Most Recent Age (years)															
15–24	1.1	(0.85–1.4)	1.2	(0.89–1.5)	0.267	1.1	(0.84–1.4)	1.2	(0.86–1.5)	0.347	1.0	(0.53–1.9)	1.2	(0.55–2.5)	0.691
25–34	2.7	(2.1–3.3)	3.8	(2.9–5.1)	<0.001	2.7	(2.1–3.4)	3.5	(2.5–4.8)	<0.001	3.6	(1.8–7.0)	7.6	(3.3–18)	<0.001
35–44	6.8	(5.4–8.5)	10	(7.4–14)	<0.001	6.9	(5.4–8.9)	9.7	(7.0–14)	<0.001	4.8	(2.6–9.2)	9.1	(4.0–22)	<0.001
45–54	11	(8.4–13)	14	(11–20)	<0.001	9.4	(7.3–12)	14	(10–20)	<0.001	6.8	(3.7–12)	16	(7.0–38)	<0.001
55–75	21	(16–26)	23	(17–32)	<0.001	14	(11–19)	22	(15–32)	<0.001	12	(6.7–21)	31	(14–72)	<0.001
Sex															
Male	0.95	(0.85–1.1)	1.1	(0.97–1.3)	0.125	0.90	(0.79–1.0)	1.1	(0.93–1.3)	0.300	1.0	(0.75–1.4)	1.2	(0.86–1.8)	0.256
Disease Status															
Severe RHD	1.9	(1.7–2.2)	0.18	(0.14–0.23)	<0.001	1.6	(1.4–1.9)	0.20	(0.15–0.26)	<0.001	1.1	(0.68–1.7)	0.08	(0.04–0.17)	<0.001
Mild/Moderate RHD	1.3	(1.1–1.5)	0.45	(0.37–0.56)	<0.001	1.5	(1.2–1.7)	0.49	(0.39–0.61)	<0.001	1.1	(0.64–1.8)	0.22	(0.11–0.44)	<0.001
Vital Status															
Dead	2.4	(2.0–2.8)	1.8	(1.5–2.3)	<0.001	3.1	(2.6–3.7)	1.8	(1.5–2.3)	<0.001	1.7	(0.97–3.2)	1.2	(0.62–2.4)	0.619
State of Residence															
SA	5.7	(4.1–8.1)	2.1	(1.3–3.3)	0.002	2.1	(1.3–3.3)	1.6	(0.92–2.8)	0.097	6.6	(3.0–16)	8.9	(3.4–25)	<0.001
QLD	1.8	(1.6–2.0)	0.87	(0.7–1.0)	0.131	0.85	(0.73–0.99)	0.77	(0.62–0.96)	0.018	1.6	(1.0–2.6)	2.6	(1.4–5.0)	0.004
WA	1.7	(1.4–2.0)	1.1	(0.87–1.4)	0.421	0.83	(0.68–1.0)	0.81	(0.62–1.1)	0.110	21	(8.1–73)	30	(9.7–118)	<0.001
Last Diagnosis Since Register Establishment (years)															
2+	3.8	(3.2–4.6)	2.3	(1.9–3.0)	<0.001	3.2	(2.5–4.0)	2.9	(2.2–3.9)	<0.001	1.8	(1.2–2.7)	1.2	(0.74–1.8)	0.529
Comorbidity															
yes	2.7	(2.4–3.0)	1.3	(1.1–1.5)	0.008	2.6	(2.3–3.0)	1.3	(1.1–1.5)	0.011	2.0	(1.5–2.7)	1.6	(0.97–2.5)	0.061
Drug/Alcohol Abuse															
yes	1.7	(1.5–1.9)	1.3	(1.1–1.5)	0.001	2.6	(2.3–3.0)	1.3	(1.1–1.5)	0.004	1.8	(1.3–2.6)	1.6	(1.1–2.4)	0.029
Remoteness (ARIA)															
Inner Regional	6.5	(5.5–7.6)	2.1	(1.6–2.8)	<0.001	2.7	(2.1–3.6)	2.5	(1.7–3.7)	<0.001	2.1	(1.2–3.6)	1.1	(0.48–2.3)	0.863
Outer Regional	1.7	(1.5–2.0)	1.0	(0.9–1.3)	0.736	1.3	(1.1–1.5)	1.1	(0.85–1.4)	0.515	1.3	(0.71–2.4)	0.82	(0.38–1.7)	0.600
Socioeconomic Status															
Quintile III	0.41	(0.33–0.52)	0.71	(0.53–0.95)	0.022	0.68	(0.53–0.87)	1.1	(0.83–1.5)	0.465	0.42	(0.26–0.68)	0.44	(0.25–0.76)	0.003
Quintile IV	0.51	(0.40–0.66)	0.75	(0.55–1.0)	0.061	0.63	(0.5–0.79)	1.1	(0.80–1.5)	0.521	0.52	(0.31–0.85)	0.45	(0.25–0.79)	0.006
Quintile V	0.23	(0.19–0.28)	0.58	(0.44–0.75)	<0.001	0.52	(0.43–0.64)	0.86	(0.64–1.2)	0.344	0.28	(0.18–0.43)	0.29	(0.17–0.49)	<0.001

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
