# Peer review of "Case Ascertainment on Australian Registers for Acute Rheumatic Fever and Rheumatic Heart Disease"

_ijerph, 2020, doi:10.3390/ijerph17155505_

Round 1

Reviewer 1 Report

The authors are to be commended for bringing this complex data together. 

This paper is quite complex, and data is dense. I find interest as i work in one of the jurisdictions discussed- however i am not sure if there will be generalised interest world-wide about the details of some of the strange structures and boundaries in Australia.

I found the paper hard to read (but got there in the end) because of the way it is presented.  4 objectives are outlined clearly in the introduction section and there is a nice graph which helps in understanding this.  However, results are not presented in a systematic way following these objectives.

Discussion

Of interest is data to support a discussion about usefulness of a centralised register.  How good is the data?  How should it be interpreted and used?   How could the register work better?  How does the register work with the health systems, and how do health systems work with the register?

In my simplistic thinking the register has 2 possible uses 1. to provide data about the extent and distribution and demographics of the disease to enable planning and resource allocation.   and 2. To be useful in real time in helping monitor care being provided to individual cases ie to help in reminder systems for secondary preventative treatment etc.   Could discussion focus on this with some thoughts about what the deficiencies identified mean, implications and possible remedies?

There may be other register uses - but i would be more interested in a discussion that focused more on what this data matching research could say about the both the hospital system and the register system.   For example, this data shows that children who live in the north of the country and who are Indigenous are best represented by the register.  This is good because these are the people who will most benefit and are most at risk of progression of disease to severe RHD and where focus of activity will have most benefit.

Some reasons for data not being in both systems could be discussed, for example the location of cardiac surgery is usually in metropolitan hospitals far from location of may of the people with RHD.

Do the authors advocate for a more active surveillance system?  Should the register  programs become more integrated with health systems?   Are there problems with individuals crossing borders and being counted twice or not at all?   

I read the references provided which purport to say WHO recommends a centralised register system.  Either there are other references which could be cited, or this should be changed because these references do not support that claim.  In fact WHO calls for ARF RHD programs to be developed for countries to estimate burden of disease, and to do this with programs integrated and delivered through existing health infrastructure, avoiding the establishment of new administrative or delivery frameworks.   

There are a few minor mistakes I noted .

Line 50 withthe

Line 137 using

Table 4 legend  -   recoding or recording?

Line 277 should this be inclusion or exclusion.

Author Response

The authors are to be commended for bringing this complex data together. 

This paper is quite complex, and data is dense. I find interest as i work in one of the jurisdictions discussed- however i am not sure if there will be generalised interest world-wide about the details of some of the strange structures and boundaries in Australia.

I found the paper hard to read (but got there in the end) because of the way it is presented.  4 objectives are outlined clearly in the introduction section and there is a nice graph which helps in understanding this.  However, results are not presented in a systematic way following these objectives.

Response: In order to present the results in a more systemic way we have now explicitly stated which paragraphs in the Results section address which objective.

Discussion

Of interest is data to support a discussion about usefulness of a centralised register.  How good is the data?  How should it be interpreted and used?   How could the register work better?  How does the register work with the health systems, and how do health systems work with the register?

In my simplistic thinking the register has 2 possible uses 1. to provide data about the extent and distribution and demographics of the disease to enable planning and resource allocation.   and 2. To be useful in real time in helping monitor care being provided to individual cases ie to help in reminder systems for secondary preventative treatment etc.   Could discussion focus on this with some thoughts about what the deficiencies identified mean, implications and possible remedies?

Response: We have included a paragraph explicitly stating the benefits of establishing a central, national register. It reads: The observed limitations in case ascertainment on the ARF/RHD registers and their likely causes point to the benefits of establishing a central, national ARF/RHD register based on automated notification and data management processes to achieve both more accurate epidemiological monitoring and more consistent real-time patient care including across jurisdictional boundaries. A short reference has also been added to the conclusion.

There may be other register uses - but i would be more interested in a discussion that focused more on what this data matching research could say about the both the hospital system and the register system.   For example, this data shows that children who live in the north of the country and who are Indigenous are best represented by the register.  This is good because these are the people who will most benefit and are most at risk of progression of disease to severe RHD and where focus of activity will have most benefit.

Response: We have added the points raised to the discussion that we had already provided on likely reasons for and characteristics of the focus of the registers on younger Indigenous populations residing in the North of Australia. The paragraph now reads: Operationally, the ARF/RHD registers face challenges such as limitations in resourcing with regard to infrastructure and staff. These operational constraints necessitate focusing on population groups and geographical areas where active case finding will likely identify a relatively large number of cases. It seems likely that such practical and clinically pertinent considerations have influenced the large representation of younger, Indigenous and Northern Australian cases recorded on ARF/RHD registers.

Some reasons for data not being in both systems could be discussed, for example the location of cardiac surgery is usually in metropolitan hospitals far from location of may of the people with RHD.

Response: Theoretically, geographic distance should not be a hurdle to data follow-up and accuracy, but we have added patient transfer as a possible reason for the observed gaps.

Do the authors advocate for a more active surveillance system?  Should the register  programs become more integrated with health systems?   Are there problems with individuals crossing borders and being counted twice or not at all?   

Response: Our main advocacy point lies in arguing for more integrated and automated case ascertainment. We have now also explicitly stated that this should ideally be implemented as a central national register (see discussion above). This is summarized in the conclusion as: Furthermore, moving towards more integrated and automated systems, ideally implemented as a central, national ARF/RHD register, has the potential to improve communication and cooperation between the registers and health care services, minimise the workload of clinicians and the double handling of data and thus increase case ascertainment and data accuracy.

I read the references provided which purport to say WHO recommends a centralised register system.  Either there are other references which could be cited, or this should be changed because these references do not support that claim.  In fact WHO calls for ARF RHD programs to be developed for countries to estimate burden of disease, and to do this with programs integrated and delivered through existing health infrastructure, avoiding the establishment of new administrative or delivery frameworks.   

Response: We have now cited the exact wording from the WHO resolution: “adequate monitoring and surveillance, as an integrated component of national health systems responses”.

There are a few minor mistakes I noted.

Line 50 withthe

Line 137 using

Table 4 legend  -   recoding or recording?

Line 277 should this be inclusion or exclusion.

Response: These have been addressed.

Submission Date

22 June 2020

Date of this review

07 Jul 2020 02:01:36

Reviewer 2 Report

Overall, I think this is an important work with good data to inform Australian ARF/RHD case tracking and improve current register systems. It also helps highlight which cases are being left out of the register and the regions with less reporting. I have a few comments/clarifications about the methodology, and think that the results/discussion sections could be improved. Please see my specific comments below or in the attached word document:

Consider including in the abstract and intro that this study includes both pediatric and adult populations. The first line of the intro may lead readers to believe it is pediatric-specific.

Line 49-51: For those unfamiliar with Australia’s geography, how many states/territories are there, and how much is covered by the Northern, Western, South and Queenland regions? And what is NSW? That abbreviation was never defined.

Lines 64: “hospitalisation is required for all ARF episodes as soon as possible after symptom onset.(10) RHD cases can be expected to be hospitalised for case management, especially for RHD-associated complications.”

Is it true that most ARF and RHD cases are expected to be hospitalized? What if mild RHD is picked up in the community based on a murmur and the patient is minimally symptomatic or asymptomatic. I think it may be a jump to conclude that ARF/RHD register records should appear in hospital databases.

Line 94: Why only include cases in either data source 1 year following establishment of the register? Was this to allow time for it to be implemented/functional? Figure 2 suggests that nearly 2,000 were excluded for this reason (“outside study period”).

Line 98: why did you exclude those diagnosed outside their own jurisdiction? Did you assume they would not be hospitalized in the corresponding region? If all new diagnoses are supposed to be hospitalized at time of diagnosis, then wouldn’t those individuals likely be hospitalized in the jurisdiction where they were diagnosed? You may answer this in your limitations section, but would be helpful to briefly describe in methods, as well.

How did you link patients between hospital records and registers based on personal identifying information? Is this the mention of “linked administrative data?” If so, please be more clear as to how this worked, as it is an important part of the methods. If not, then new ARF/RHD by hospitalizations are likely to be grossly overestimated, as one case could be counted multiple times. You mentioned the ARF admissions needing to be >90 from prior, but what about RHD patients? Example: a case diagnosed with RHD at first hospitalization is entered in the register at that time, then is readmitted over the years for recurrent issues, like CHF or procedures, and counted multiple times as hospital RHD cases.

Line 175: “NT had the highest proportion of individuals appearing in both data sources.” This seems contradictory to the previously mentioned results, that Indigenous cases, with the lowest SES, were from the NT, and lowest SES were not found in hospital data. This is likely because NT had the largest number of cases compared to other states, and therefore can be misleading.

Consider editing paragraph for non-indigenous cases (starting line 178): “The cases appearing in the hospital data only were mostly from the most socioeconomically advantaged areas, while cases appearing on both the register and hospital data or the register record only mainly resided in the most disadvantaged areas (p<0.001)...” The words “mostly,” “mainly,” and “more” are qualitative, can you replace with quantitative data?

Figure 2: How did you get from initial 45,300 cases to 43,230? I was surprised to see so many cases >60 years of age in the registry, accounting for >1/2 of all register cases. Any thoughts/explanation for this?

Section 3.2.1 and 3.2.2: If you are not doing statistical analysis to compare the register cases by region, then you cannot make statements that one area was higher than the other, without p-values to show it is actually significant. You can state individual regions percentages and discuss trends.

Figure 3: The figure is blurry – assume this will be updated for final submission. The Figure is confusing with percentages and cases mixed. Consider just presenting the percentages and removing the case number to simplify.

Line 229-230: “Older age, comorbidities, drugs/alcohol abuse and disease severity but not sex were determinants of register inclusion.” Was this on univariate or multivariate analysis? Would be helpful to be clear in this paragraph, which analysis you are describing for each result.

Line 277: “We found that non-Indigenous status remains a significant predictor of register inclusion.” I thought the data showed the opposite? That non-indigenous status was a significant predictor of exclusion from the register and inclusion in the hospital data only, and as you stated at the beginning of the discussion: 76% of non-indigenous cases were missing from the register.

Your discussion does a great job of highlighting regional differences in inclusion into the register, starting with line 287. This includes differences in mandated reporting, timeline of register development, and logistic issues with reporting systems. However, I think you could expand on results that you describe in the first 3 paragraphs of the discussion, and generate hypotheses as to why certain demographic/clinical features (age, disease severity, etc) or Indigenous status was predictive of these differences. (you do this a bit in lines 306-316).

Your conclusion does not include any mention of differences in populations (i.e. Indigenous status), which is a major component of your results. Consider adding.

Lastly, how does this study help inform the greater RHD research/clinical community? Can you draw conclusions that help influence other regions or is this only applicable to Australia?

Author Response

Overall, I think this is an important work with good data to inform Australian ARF/RHD case tracking and improve current register systems. It also helps highlight which cases are being left out of the register and the regions with less reporting. I have a few comments/clarifications about the methodology, and think that the results/discussion sections could be improved. Please see my specific comments below or in the attached word document:

Consider including in the abstract and intro that this study includes both pediatric and adult populations. The first line of the intro may lead readers to believe it is pediatric-specific.

Response: We have inserted that the study contains both paediatric and adult populations in line 92.

Line 49-51: For those unfamiliar with Australia’s geography, how many states/territories are there, and how much is covered by the Northern, Western, South and Queenland regions? And what is NSW? That abbreviation was never defined.

Response: NSW is New South Wales. It is defined the first time it is used in line 54. We have added that the five jurisdictions are home to 86% of the Australian Indigenous population.

Lines 64: “hospitalisation is required for all ARF episodes as soon as possible after symptom onset.(10) RHD cases can be expected to be hospitalised for case management, especially for RHD-associated complications.”

Is it true that most ARF and RHD cases are expected to be hospitalized? What if mild RHD is picked up in the community based on a murmur and the patient is minimally symptomatic or asymptomatic. I think it may be a jump to conclude that ARF/RHD register records should appear in hospital databases.
Response: As mentioned in the Discussion (around line 330), our data does not include primary health care records. Therefore, our estimates are likely underestimates of the true gap in case ascertainment through the ARF/RHD registers. With progressing disease, we would expect patents to be treated in hospital for management and complications. In that sense, any records identified only through the hospital data are likely more advanced disease (unless incidental finding). In any case, at a more advanced stage, it seems even more imperative that cases should be included on ARF/RHD registers.

Line 94: Why only include cases in either data source 1 year following establishment of the register? Was this to allow time for it to be implemented/functional? Figure 2 suggests that nearly 2,000 were excluded for this reason (“outside study period”).

Response: “Outside study period” also includes cases in the parent ERASE dataset that preceded register establishment in the first place. In addition, we have allowed for a period for registers to become operational. We know from discussion with register managers that the set-up periods were staggered and it is likely that more cases were missed in the very beginning for operational reasons that do not persist in the future.

Line 98: why did you exclude those diagnosed outside their own jurisdiction? Did you assume they would not be hospitalized in the corresponding region? If all new diagnoses are supposed to be hospitalized at time of diagnosis, then wouldn’t those individuals likely be hospitalized in the jurisdiction where they were diagnosed? You may answer this in your limitations section, but would be helpful to briefly describe in methods, as well.

Response: Hospitalization is only recommended for ARF upon diagnosis, not for RHD. We do not have cross-jurisdictionally linked data except for NT and SA. Therefore, we cannot tell from the data whether a case was hospitalised in another jurisdiction after their diagnosis. We exclude these cases, because we do not have follow-up for them. We have added some clarification in line 94.

How did you link patients between hospital records and registers based on personal identifying information? Is this the mention of “linked administrative data?” If so, please be more clear as to how this worked, as it is an important part of the methods. If not, then new ARF/RHD by hospitalizations are likely to be grossly overestimated, as one case could be counted multiple times.

Response: A clarifying sentence has been added in line 72: Linked administrative data allows for the follow-up of cases across health system contacts and facilities over time, avoiding overestimation of diagnoses and caseloads.

You mentioned the ARF admissions needing to be >90 from prior, but what about RHD patients? Example: a case diagnosed with RHD at first hospitalization is entered in the register at that time, then is readmitted over the years for recurrent issues, like CHF or procedures, and counted multiple times as hospital RHD cases.

Response: As per previous response, since we have access to linked data, we would count this person only as one RHD case.

Line 175: “NT had the highest proportion of individuals appearing in both data sources.” This seems contradictory to the previously mentioned results, that Indigenous cases, with the lowest SES, were from the NT, and lowest SES were not found in hospital data. This is likely because NT had the largest number of cases compared to other states, and therefore can be misleading.

Response: Yes, this is likely because of overall greater case numbers in NT. We have added some clarification as suggested.

Consider editing paragraph for non-indigenous cases (starting line 178): “The cases appearing in the hospital data only were mostly from the most socioeconomically advantaged areas, while cases appearing on both the register and hospital data or the register record only mainly resided in the most disadvantaged areas (p<0.001)...” The words “mostly,” “mainly,” and “more” are qualitative, can you replace with quantitative data?

Response: This has now been resolved.

Figure 2: How did you get from initial 45,300 cases to 43,230? I was surprised to see so many cases >60 years of age in the registry, accounting for >1/2 of all register cases. Any thoughts/explanation for this?

Response: This change in sample size from 45,300 to 43,300 was due to changes made to the parent data set to exclude those with congenital heart disease.

Section 3.2.1 and 3.2.2: If you are not doing statistical analysis to compare the register cases by region, then you cannot make statements that one area was higher than the other, without p-values to show it is actually significant. You can state individual regions percentages and discuss trends.

Response: This has now been resolved.

Figure 3: The figure is blurry – assume this will be updated for final submission. The Figure is confusing with percentages and cases mixed. Consider just presenting the percentages and removing the case number to simplify.

Response: We have removed the absolute numbers as suggested.

Line 229-230: “Older age, comorbidities, drugs/alcohol abuse and disease severity but not sex were determinants of register inclusion.” Was this on univariate or multivariate analysis? Would be helpful to be clear in this paragraph, which analysis you are describing for each result.

Response: We have clarified that all results in Table 4 are based on multivariate analyses.

Line 277: “We found that non-Indigenous status remains a significant predictor of register inclusion.” I thought the data showed the opposite? That non-indigenous status was a significant predictor of exclusion from the register and inclusion in the hospital data only, and as you stated at the beginning of the discussion: 76% of non-indigenous cases were missing from the register.

Response: We apologize for the mistake. This has been resolved. It now says: Indigenous status remains a significant predictor of register inclusion.

Your discussion does a great job of highlighting regional differences in inclusion into the register, starting with line 287. This includes differences in mandated reporting, timeline of register development, and logistic issues with reporting systems. However, I think you could expand on results that you describe in the first 3 paragraphs of the discussion, and generate hypotheses as to why certain demographic/clinical features (age, disease severity, etc) or Indigenous status was predictive of these differences. (you do this a bit in lines 306-316).

Response: We have added to the explanation now saying that: Operationally, the ARF/RHD registers face challenges such as limitations in resourcing with regard to infrastructure and staff. These operational constraints necessitate focusing on population groups and geographical areas where active case finding will likely identify a relatively large number of cases. It seems likely that such practical and clinically pertinent considerations have influenced the large representation of younger, Indigenous and Northern Australian cases recorded on ARF/RHD registers.

There is no specific mandate to prioritize certain demographic and clinical subgroups for register inclusion, but as explained practical considerations have meant that younger indigenous patients have (rightly so) been the focus of surveillance efforts.

Your conclusion does not include any mention of differences in populations (i.e. Indigenous status), which is a major component of your results. Consider adding.

Response: This has been added, as suggested. We do not feel that it requires a lengthy summary, because the clinically pertinent focus on younger Indigenous populations groups is well-documented.

Lastly, how does this study help inform the greater RHD research/clinical community? Can you draw conclusions that help influence other regions or is this only applicable to Australia?

Response: We have added that this study demonstrates the need for sophisticated monitoring and surveillance systems in the global effort to reduce the burden of ARF and RHD.

Reviewer 3 Report

The article is very interesting and claim to the attention to the authorities to the healthy problems caused by S. pyogenes infections, like rheumatic fever, Rheumatic Heart disease and impetigo, that occurs more frequently in indigenous populations in Australia.

Author Response

Comments and Suggestions for Authors

The article is very interesting and claim to the attention to the authorities to the healthy problems caused by S. pyogenes infections, like rheumatic fever, Rheumatic Heart disease and impetigo, that occurs more frequently in indigenous populations in Australia.

Response: We appreciate that you found our manuscript interesting.

Round 2

Reviewer 2 Report

Your edits in response to my initial comments were satisfactory and I have no further comments about the manuscript.